# Effect of Nano-Selenium on Nutritional Quality of Cowpea and Response of ABCC Transporter Family

**DOI:** 10.3390/molecules28031398

**Published:** 2023-02-01

**Authors:** Li Li, Yuzhou Xiong, Yuan Wang, Shuai Wu, Chunmei Xiao, Shiyan Wang, Shuiyuan Cheng, Hua Cheng

**Affiliations:** 1School of Modern Industry for Selenium Science and Engineering, Wuhan Polytechnic University, Wuhan 430023, China; 2National R&D Center for Se-rich Agricultural Products Processing, Wuhan Polytechnic University, Wuhan 430023, China

**Keywords:** biological nano selenium, nutritional composition, antioxidant stress, organic selenium, ABC transporter family

## Abstract

It is an important way for healthy Selenium (Se) supplement to transform exogenous Se into organic Se through crops. In the present study, *Vigna unguiculata* was selected as a test material and sprayed with biological nano selenium (SeNPs) and Na_2_SeO_3_, and its nutrient composition, antioxidant capacity, total Se and organic Se content were determined, respectively. Further, the response of ABC transporter family members in cowpea to different exogenous Se treatments was analyzed by transcriptome sequencing combined with different Se forms. The results show that the soluble protein content of cowpea increased after twice Se treatment. SeNPs treatment increased the content of cellulose in cowpea pods. Na_2_SeO_3_ treatment increased the content of vitamin C (Vc) in cowpea pods. Se treatments could significantly increase the activities of Peroxidase (POD), polyphenol oxidase (PPO) and catalase (CAT) in cowpea pods and effectively maintain the activity of Superoxide dismutase (SOD). SeNPs can reduce the content of malondialdehyde (MDA) in pods. After Se treatment, cowpea pods showed a dose-effect relationship on the absorption and accumulation of total Se, and Na_2_SeO_3_ treatment had a better effect on the increase of total Se content in cowpea pods. After treatment with SeNPs and Na_2_SeO_3_, the Se species detected in cowpea pods was mainly SeMet, followed by MeSeCys. Inorganic Se can only be detected in the high concentration treatment group. Analysis of transcriptome data of cowpea treated with Se showed that ABC transporters could play an active role in response to Se stress and Se absorption, among which ABCB, ABCC and ABCG subfamilies played a major role in Se absorption and transportation in cowpea. Further analysis by weighted gene co-expression network analysis (WGCNA) showed that the content of organic Se in cowpea treated with high concentration of SeNPs was significantly and positively correlated with the expression level of three transporters ABCC11, ABCC13 and ABCC10, which means that the ABCC subfamily may be more involved in the transmembrane transport of organic Se in cells.

## 1. Introduction

Selenium (Se) is one of the essential trace elements for human body and has very important physiological functions [1]. At present, a total of 25 seleno-proteins have been identified in human proteome, usually oxidoreductases, including SeCys as catalytic residues [2]. Human Se deficiency is associated with thyroid dysfunction, irreversible brain damage, peripheral vascular disease, chronic and degenerative osteoarthropathy (Kashin Beck disease), impaired immune response to viral infection, male infertility, pre-eclampsia in women, heart disease and increased risk of several cancers [3]. In addition, excessive selenium intake can also lead to poisoning. Recent studies have shown that even subtoxic doses may have negative effects, such as an increased risk of type II diabetes [1]. People living in the USA and Canada usually do not have problems related to Se deficiency. On the contrary, people living in China, New Zealand, parts of Europe and occasionally Russia have insufficient intake of micronutrients due to the low Se content in soil and food [4]. According to the research of the Institute of Medicine of the USA National Academy, the recommended dietary intake of Se for adults is 55 µg/day, while the upper tolerable intake for adults is 400 µg/day [5]. Selenate (SeO_4_^2−^) and Selenite (SeO_3_^2−^) are the two major forms of bioavailable Se in soil, with selenate predominating in oxic soils and selenite predominating in anoxic soils [6]. Organic seleno-compounds such as seleno-amino acids are also present in significant concentrations in some soils and can be imported by plant roots [7]. In its early stage, the cultivation technology of increasing Se content in plants is called Se-biofortification, and has been developed for several plants, including trees, crops and vegetables [8,9,10,11,12].

*Vigna unguiculata* (Linn.) is an important leguminous plant with rich nutritional value and adaptability [13]. Due to its resistance to water loss, it is suitable to hot arid regions [14]. Cowpea is also used as a dietary protein and mineral source as a major vegetable crop in the diets of developing countries in Africa, Latin America and Asia [15]. Se concentrations in plants grown under acidic and weathering conditions (such as tropical soils) are often very low. It is estimated that more than 1 billion people worldwide are Se deficient [13,16]. Therefore, supplementing Se through crops may be an important strategy to increase the daily intake of Se by humans and animals [17]. Although Se has important nutritional value and is beneficial to plant growth, the range between Se deficiency and poisoning is very narrow [16]. High concentrations of Se can affect the normal growth of crops, such as cowpea [18]. High doses of Se in plants can cause stress response and increase of reactive oxygen species, and finally cause crop damage. The rise of antioxidative enzymes such as peroxidase (POD), superoxide dismutase (SOD), and catalase (CAT) in response to high levels of heavy metal exposure also indicates abiotic stress in plants [13,17].

The existing studies indicate that ABC transporters may be involved in the process of Se absorption and transport in plants [19,20]. The ABC transporter is one of the largest and oldest protein families found at present, which widely exists in eukaryotes and prokaryotes. According to the phylogenetic analysis of ABC transporters, the similarity of nucleotide binding region sequences and the organizational form of domains, the family can be divided into eight subfamilies, ABCA to ABCG and ABCI [21]. Studies have shown that phosphate transporters and silicon transporters participate in the absorption of Na_2_SeO_3_ by plants [22,23]. The rice phosphorus transporter *OsPT2* is mainly expressed in the root, which can promote the effective absorption of Se in crops. After the Se absorbed from the outside is converted into organic Se, it is transported to the upper part of the root. Overexpression of *OsPT2* gene can improve the tolerance of crops to Se [19]. The expression changes of 28 ABC transporters in *Lolium perenne* may be related to the accumulation of selenides after selenate saline culture [20]. After *Oryza sativa* was treated with SeNPs and Na_2_SeO_3_, seven differentially expressed ABC transporters were screened, of which five were up-regulated and two were down-regulated, indicating that ABC transporters may also be involved in the absorption and transport of Se by rice [24]. These results indicated that ABC transporters are closely related to Se accumulation in plants.

In present study, cowpeas were treated with SeNPs and Na_2_SeO_3_ to explore the effects of Se on physiological indexes, antioxidant capacity, total Se content and Se species of cowpeas. Further analysis of transcriptome data revealed that ABCC transporter family may participate in the transport of Se and the mechanism of organic Se transformation. The research results are expected to provide a basis for the establishment of efficient Se enriched cultivation method of cowpea, determine the best Se concentration, times of Se application and harvest time of Se enriched cowpea, and provide new insights for leguminous plants to cope with Se stress and the role of ABC transporter in Se transport of legume.

## 2. Results

### 2.1. Physiological Indexes of Cowpea Pods

After SeNPs treatment, the content of soluble sugar in cowpea pods decreased first and then increased with the increase of concentration, and there was no significant difference compared with the control group (Figure 1a). Compared with the control group, 0.1–2.5 mmol/L treatment reduced the soluble sugar content of pods, and the lowest concentration of 1.0 mmol/L was 11.64 mg/g. Compared with the control group, the content of soluble sugar increased at 3.0 mmol/L concentration, and the content of soluble sugar was 14.15 mg/g. After Na_2_SeO_3_ spraying treatment, the content of soluble sugar in cowpea pods had no significant difference compared with the control group and was lower than the control group. The lowest value was 10.07 mg/g under 0.1 mmol/L treatment concentration, and the highest value was 12.24 mg/g under 3.0 mmol/L.

After SeNPs treatment, the content of soluble protein in cowpea pods was significantly increased compared with the control group: the content was 0.53 mg/g, 0.44 mg/g, 0.52 mg/g, 0.48 mg/g, 0.44 mg/g, respectively, which was 253.3%, 193.3%, 246.7%, 220.0%, 193.3% higher than the control group. After Na_2_SeO_3_ treatment, the soluble protein content in cowpea pods increased compared with the control group. The concentrations of 0.5 mol/L and 3.0 mmol/L showed significant difference, with the contents of 0.32 mg/g and 0.53 mg/g, respectively, which were 113.3% and 253.3% higher than those of the control group (Figure 1b).

After the cowpea leaves were treated with SeNPs, the cellulose content in the pod was higher than that in the control group. The concentration of 1.0–2.5 mmol/L was the most significant, and the cellulose content was 5.62 mg/g, 6.56 mg/g, 6.56 mg/g and 6.51 mg/g, respectively. After 3.0 mmol/L Na_2_SeO_3_ treatment, the cellulose content in cowpea pods was significantly increased compared with the control group, the content was 5.71 mg/g, and the increase was 24.5%. There was no significant difference in the other Na_2_SeO_3_ treatment group compared with the control (Figure 1c).

After spraying cowpea leaves with SeNPs, the Vc content in the pod had no significant change compared with the control group. After spraying cowpea with Na_2_SeO_3_, the content of Vc in cowpea pods increased first and then decreased with the increase of Na_2_SeO_3_ concentration, and both were higher than the control group, but there was no significant difference (Figure 1d).

### 2.2. Effect of Se on Antioxidant System

After SeNPs treatment, the content of malondialdehyde (MDA) in cowpea pods was significantly lower than that in the control group (Figure 2a). Under the treatment concentration of 3.0 mmol/L, the content of MDA reached a minimum of 0.23 mg/g, with a decrease of 42.5%. After Na_2_SeO_3_ treatment, the content of MDA in cowpea pods of all treatment groups increased, and the content of MDA in cowpea pods was significantly increased under 0.1 mmol/L treatment, which was 41.0% higher than that of the control group.

After cowpea was treated with 0.1 mmol/L, 0.5 mmol/L and 2.5 mmol/L SeNPs, the content of H_2_O_2_ in the pod was significantly higher than that of the control group, which was increased by 195%, 78%, 122%, respectively. After cowpea was treated with 3.0 mmol/L SeNPs, the content of hydrogen peroxide in the pod was slightly lower than that in the control group. After Na_2_SeO_3_ treatment, the content of hydrogen peroxide in pods first increased and then decreased, and all treatments were higher than the control group, among which 0.5–1.0 mmol/L concentration had a significant increase (Figure 2b).

After SeNPs treatment, POD activity in cowpea pods increased first and then decreased. After 0.1–2.5 mmol/L treatment, the POD activity was improved compared with the control group. After 0.1–1.0 mmol/L treatment, the POD activity increased by 56.6%, 91.4% and 47.9%, respectively, compared with the control group. However, the POD activity of the treatment group with 3.0 mmol/L concentration was lower than that of the control group. The results showed that the high concentration of SeNPs would reduce the POD activity of cowpea. After Na_2_SeO_3_ treatment, the activity of POD in cowpea pods was improved compared with the control group. Except that there was no significant difference at the concentration of 0.5 mmol/L, the activity of POD in other treatment groups was increased by 126.2%, 39.1%, 91.4%, 63.0% and 91.4%, respectively, compared with the control group (Figure 2c).

After 0.1 mmol/L Na_2_SeO_3_ treatment, the SOD activity in cowpea pods was significantly lower than that in the control group, which was 91% of the control group. After treatment with 0.1 mmol/L and 2.5 mmol/L SeNPs, the SOD activity in pods decreased compared with the control group, which was 97% and 96% of the control group, respectively, with no significant difference (Figure 2d).

With the increase of SeNPs concentration, the CAT activity in cowpea pods decreased first and then increased. After 0.1–0.5 mmol/L treatment, the CAT activity in cowpea pods was lower than that in the control group. When the concentration was 2.5 and 3 mmol/L, the CAT activity was significantly higher than that of the control group, increased by 239.2% and 147.9%, respectively. After Na_2_SeO_3_ treatment, the CAT activity in the 0.5 and 1.0 mmol/L treatment groups was significantly increased compared with the control group, with the increase amplitude of 81.3% and 62.1%, respectively (Figure 2e).

The SeNPs treatment can significantly improve the PPO enzyme activity of cowpea. When the concentration of the treatment is greater than 0.5 mmol/L, the PPO activity is increased by 45.45%, 198.73%, 238.46% and 251.44%, respectively, compared with the control. In the Na_2_SeO_3_ treatment group, only 1 and 2.5 mmol/L treatment groups had significant improvement, which were 36.33% and 204.89% higher than the control, respectively (Figure 2f).

### 2.3. Effect of Exogenous Se Treatment on Total Se Content in Cowpea Pods

Figure 3 shows that the Se content in cowpea pods increases significantly with the increase of exogenous Se concentration. When the concentration of SeNPs is more than 1.0 mmol/L, the Se content in the pod increases significantly. When the treatment concentration is 3.0 mmol/L, the Se content in the pods reaches the maximum of 12.81 μg/g. At low concentration (<0.5 mmol/L), the effect of Na_2_SeO_3_ on the increase of Se content in cowpea pods was not as good as that of SeNPs. When the concentration of exogenous Se is more than 1.0 mmol/L, the effect of Na_2_SeO_3_ is more obvious. Compared with the control, the Se accumulation in cowpea pods was significantly increased by 3.22 times, 4.47 times and 8.00 times after the application of 0.5~3.0 mmol/L Na_2_SeO_3_, and the difference between the treatment groups was significant.

Under the condition of the same Se source and the same treatment concentration, different picking time has a greater impact on the total Se content in cowpea pods. Among them, in the 0.5 mmol/L treatment group, the total Se content in the pods picked the second time was slightly lower than that in the first time, while in the high concentration (2.5 mmol/L) treatment group, the total Se content in the pods picked the second time was higher than that in the first time. In the 2.5 mmol/L SeNPs treatment group, the Se content in the second harvest pods was significantly higher than that in the first harvest, about 2.16 times (Figure 4a).

Figure 4b shows that different Se application times will affect the total Se content of cowpea. The total Se content of soybean pods with the second application of Se was higher than that with the first application of Se. Among them, the treatment concentration of 0.5 mmol/L has a significant increase effect, which is about 3.05 times higher. In the 0.5 mmol/L Na_2_SeO_3_ treatment group, the total Se content in the pods with the second application of Se was slightly higher than that with the first application of Se, while the total Se content in the pods with the second application of Se in the 2.5 mmol/L treatment group decreased significantly (Figure 4b).

### 2.4. Effects of Se Application on the Forms and Contents of Se in Cowpea Pods

In all treatments, SeMet was the main form of Se in cowpea pods, accounting for 55–100%. When the concentration of SeNPs and Na_2_SeO_3_ was 0~0.1 mmol/L, only the unique Se form, SeMet, was detected in the pods; with the increase of exogenous Se concentration, the detected Se species increased gradually. They all exist in the form of organic Se; when the concentration of exogenous Se is 3.0 mmol/L, inorganic Se (Se^4+^, Se^6+^) can be detected (Table 1).

When the concentration of SeNPs treatment is 0.5 mmol/L, the Se forms in cowpea pods are mainly MeSeCys and SeMet, accounting for 28.65% and 71.35%, respectively. When the concentration of SeNPs was 1.0 and 2.5 mmol/L, the Se in cowpea pods was mainly SeCys2, MeSeCys and SeMet, accounting for 2.17%, 26.93%, 70.90% and 4.45%, 32.31% and 63.24%, respectively. When the concentration of SeNPs treatment was 3.0 mmol/L, five Se forms were detected, including Se^4+^, Se^6+^, SeCys2, MeSeCys and SeMet, of which inorganic Se accounted for 13.88% and organic Se accounted for 86.12%. Table 1 shows that the content of various Se forms detected in cowpea pods increases with the increase of SeNPs concentration applied, and there are significant differences among the treatments.

Different from SeNPs treatment, after 0.5 mmol/L Na_2_SeO_3_ treatment, three forms of organic Se can be detected: SeCys2, MeSeCys and SeMet, which account for 1.21%, 14.82% and 83.97%, respectively. When the concentration of Na_2_SeO_3_ increases to 3.0 mmol/L, four Se forms, namely Se^4+^, SeCys2, MeSeCys and SeMet, can be detected, accounting for 3.12%, 6.28%, 18.61% and 71.99%, respectively (Table 2). The content of various Se forms detected in cowpea pods gradually increased with the increase of Na_2_SeO_3_ concentration.

### 2.5. Analysis of Transcriptome Data of Cowpea Treated with Se

The sequencing results showed that the contrast rate of all treatment groups remained at 82.99–94.89%, indicating that the reference genome assembly could meet the needs of information analysis and the reliability of later data analysis (Appendix A). A sample correlation analysis was carried out between each two of all treatment groups, the Pearson correlation coefficient calculated between all gene samples and a heat map was made. The results showed that the *p*-value value was between 0.938 and 0.995, indicating that the similarity within the sample group was very high and the number of different genes between samples was small (Appendix A). The transcriptome of the treated samples was analyzed. It was found that the relative expression value of FPKM (Log10) was between 0–1.5, indicating that the samples had high repeatability and reliability (Appendix A).

All genes were analyzed for differential expression (the screening threshold was q value <0.05 and | log2FoldChange |>1). A total of 3778 differentially expressed genes (DEGs) were screened, of which 3680 were annotated, accounting for 11.5% of the total genes. GO enrichment analysis showed that DEGs are mainly enriched in molecular function, kinase activity and DNA binding subclasses; cell components are enriched in cellular component and protein complex subclasses; and molecular biological processes are mainly enriched in response to stress, transmembrane transport and transport (Appendix A, Appendix A).

The cowpea transcriptome data were analyzed, and the transcriptome databases of *Arabidopsis thaliana*, *Nicotiana tabacum* and *O. sativa* were used for comparative annotation. A total of 31,948 genes were annotated, and the number of genes annotated by *Arabidopsis*, tobacco and rice was 17,464, 9522 and 17,733, respectively. Only 2552, 1391 and 2852 genes were compared and annotated separately by *A. thaliana*, *N. tabacum* and *O. sativa*. A total of 6328 genes were compared and annotated by the three species (Figure 5a).

Using ABC transporter as the key word, 107 genes encoding ABC transporter were screened from cowpea transcriptome data (Figure 5b). The comparison results showed that 93, 79 and 84 ABC transporter genes were annotated into the databases of arabidopsis, tobacco and rice, respectively. Among them, there were 2, 11 and 5 genes annotated by three species alone, accounting for 1.7%, 9.4% and 4.3%, respectively. The number of ABC protein genes annotated by the three species was 40, accounting for 34.2%.

The results of comparison and screening of DEGs according to different treatment groups are shown in Figure 6. Among them, the control group has the largest number of DEGs compared with the SeNPs treatment group, indicating that a large number of genes in cowpea leaves respond to Se stress after SeNPs treatment, which may play a role in promoting the absorption of SeNPs in cowpea leaves. SeNPs treatment showed that the number of DEGs between high concentration treatment groups was significantly less, while the number of DEGs between low concentration and high concentration treatment groups (N-100 vs. N-2500, N-100 vs. N-3000, N-500 vs. N-3000) was more. It is speculated that with the increase of SeNPs concentration, more genes respond to the stress of SeNPs treatment on cowpea leaves. The number of DEGs between different Se species treatments (N-1000 vs. S-1000) is also high, which may be related to the different ways of absorbing SeNPs and Na_2_SeO_3_ in leaves (Figure 6).

### 2.6. Analysis of Transcriptome ABC Transporter Gene

Seven ABC transporter subfamilies were identified in the Se treated cowpea transcriptome data, of which ABCB, ABCC, and ABCG have a large number of 32, 15, and 47, respectively. The three subfamilies account for 80.3% of the total ABC transporter genes. The number of ABCA subfamilies is 14. In cowpea ABC transporters, the three subfamilies of ABCD, ABCE, and ABCF are relatively small. No ABCH subfamily genes were detected in cowpea transcriptome (Figure 7a).

Cowpea has 11 pairs of chromosomes (2n = 22). ABC transporter genes are unevenly distributed on cowpea chromosomes. The number of ABC transporter genes on chromosome No. 3 is up to 21, accounting for 18.0%; The second is chromosome No. 7 and No. 1, which contain 17 and 16 genes, respectively. There are four ABC transporter genes distributed on chromosomes No.2 and No.11, respectively (Figure 7b).

### 2.7. Differential Expression Analysis of ABC Transporter Gene

Among the DEGs of transcriptome data, 107 ABC transporter genes were screened from cowpea pods. The ABCG transporter subfamily has 44 genes at most, followed by ABCB and ABCC transporter subfamilies, with 27 and 14 DEGs, respectively. It indicated that the ABCG, ABCB and ABCC subfamilies had an obvious response to stress after exogenous Se treatment (Figure 8).

The differential expression analysis of cowpea ABC transporter gene was conducted among groups. The expression level of the gene was based on the fold changes value. When log2FoldChange > 0, it was considered that the DEGs was up-regulated. On the contrary, when log2FoldChange < 0, it was considered that the DEGs was down-regulated (Appendix A). There was no significant difference in the number of differentially expressed ABC transporter genes among the control groups. After 1.0 mmol/L SeNPs treatment, the number of differentially expressed ABC transporter genes was the largest, with 55 up-regulated and 49 down-regulated; after 0.5 mmol/L SeNPs treatment, the number of differentially expressed ABC transporter genes was the least, and the number of up/down regulated genes was 49. The differential expression analysis of ABCB, ABCC and ABCG subfamilies showed that the DEGs in ABCC subfamily were identical. The change of Se concentration had little effect on the number of DEGs in the three subfamilies, and the three subfamilies accounted for a large proportion of differentially expressed ABC transporter genes, accounting for 77.6–77.9%. These results indicate that ABCB, ABCC and ABCG subfamilies are the main transporters family in response to Se stress.

Seventeen groups of DEGs were screened, and a total of 40 highly expressed ABC transporter differentially expressed genes were screened. The number of ABCB, ABCC and ABCG subfamilies were 9, 7 and 21 genes, respectively, accounting for 92.5% of all differentially expressed ABC transporters. These data support that these three subfamilies are the main genes for ABC transporters to cope with Se stress. After increasing the screening threshold, compared with the control, the number of differentially expressed ABC transporter genes in pods of 100 μmol/L SeNPs treatment group was the largest, of which 17 genes were up-regulated and 9 genes were down regulated, indicating that ABC transporters could respond to SeNPs stress after low concentration Se application. The number of differentially expressed ABC transporter genes decreased with the increase of SeNPs concentration, which may be related to the different reversals of cowpea leaves caused by excessive SeNPs concentration. Compared with treatment group SeNPs and Na_2_SeO_3_ at the same concentration, there were 8 up-regulated genes and 10 down-regulated genes for ABC transporter (Figure 9).

### 2.8. GO Enrichment Analysis of ABC Transporters

The GO enrichment analysis of differentially expressed ABC protein genes showed that all 40 genes were enriched to ATP binding proteins in molecular functions. In cell components, they are mainly concentrated in membrane and membrane components; in the biological process, there are mainly three subtypes of transport, transmembrane transport and transport activities. The main function of ABC transporter is to complete the transmembrane transport of proteins, lipids and other compounds by hydrolyzing ATP to generate energy, which is completely consistent with the significant enrichment analysis of ABC transporter GO function, further indicating that ABC transporter plays an important role in the absorption and transport of Se in cowpea (Figure 10).

### 2.9. WGCNA Analysis of DEGs

In order to further screen the differentially expressed ABC genes related to the absorption and transport of Se, we conducted correlation analysis between the DEGs and the contents of five Se species in cowpea pods. In the analysis, we set the power value of the module as 10, the similarity parameter as 0.75, and the minimum gene number of the module as 50 (Figure 11a). WGCNA analysis finally obtained 30 modules with similar gene expression patterns, each module contains 76–3332 genes (Figure 11b). Figure 11c shows the correlation analysis of the module characteristic genes and five forms of Se (SeCys2, MeSeCys, Se^4+^, SeMet, Se^6+^). It can be seen from the figure that White and Medium purple3 modules are positively correlated with the contents of organic Se, total Se, SeCys2, MeSeCys and SeMet, while Honeydew modules are negatively correlated with the contents of organic Se and total Se, indicating that the characteristic genes of these three modules are highly correlated with the Se accumulation of cowpea (Figure 11c).

In order to display the details of the relevant modules, the first 20 KEGG pathways of the two positive correlation modules are displayed using bubble diagrams (Figure 12). In the White module, N3001 sample has the most up-regulated genes (Figure 12a), and the genes are significantly enriched in the plant hormone signal transport, plant pathogen interaction, and nuclear cellular transport metabolic pathways. It is worth noting that *ABCC11* (Vigun05g292000) protein is enriched in the ABC transporters pathway (Figure 12c). In the Medium purple3 module, N3002 has the most up-regulated genes (Figure 12b), and the genes are significantly enriched in the biosynthesis of secondary metals, biosynthesis of amino acids, nitrogen metabolism, ether lipid metabolism and ABC transporters metabolic pathways. Among them, *ABCC13* (Vigun05g291860) and *ABCC10* (Vigun05g292884) genes are enriched in the metabolic pathway of ABC transporters (Figure 12d).

## 3. Discussion

### 3.1. Effect of Se on Physiological Indexes of Cowpea Pods

Cowpea pod is rich in protein, soluble sugar, cellulose and polyphenolic compounds, and long-term consumption can effectively supplement the human body needs nutrients [25,26,27,28]. Se is an essential trace element for human body and also a beneficial element for plant growth [11,29]. Relevant studies showed that Se application could improve the physiological indicators of leguminous plants, and exogenous Se leaf spraying could effectively increase the content of soluble protein in cowpea pods [30]. The seed soaking with 4 mg/L Na_2_SeO_3_ could significantly increase the content of soluble protein, soluble sugar and Vc in cowpea by 38%, 28% and 47%, respectively, compared with the control [31]. After selenite treatment of soybean plants at the early flowering and full flowering stages, the fat content in seeds did not change significantly, but the protein content increased significantly [32]. Treatment of peanut leaves with 20 and 40 mg/L SeNPs will increase the total sugar concentration in different branches [33]. Soaking *Vigna radiata* seeds with 0.75 mg/L Na_2_SeO_4_ will increase the concentration of soluble sugar in *V. radiata* sprouts [34].

This study showed that the two Se sources had effects on the physiological indexes of cowpea pods, and the effects of different Se sources on the nutritional components of cowpea were different. The content of soluble protein in cowpea pods treated with two kinds of Se was higher than that of the control, and the SeNPs treatment showed a significant difference. After two kinds of Se treatment, the cellulose content in cowpea pods increased compared with the control group, and 1.0–2.5 mmol/L SeNPs treatment showed a significant difference. Compared with the control group, the cellulose content in cowpea pods treated with 3.0 mmol/L Na_2_SeO_3_ was significantly increased. After Na_2_SeO_3_ treatment, the content of Vc in pods increased, but there was no significant difference. The change of physiological indexes of cowpea pods is an important factor determining the commodity attributes. In the current study, it was found that SeNPs has a greater promotion impact on the nutritional indicators of cowpea than Na_2_SeO_3_.

### 3.2. Effect of Exogenous Se Treatment on Total Se Content in Cowpea Pods

After applying exogenous Se (SeNPs or Na_2_SeO_3_), the absorption and accumulation of total Se in cowpea pods showed a dose-response relationship. The results were consistent with the research results on other leguminous plants such as *Pisum sativum*, *Astragalus sinicus* and *Medicago sativa* [35,36,37]. The results showed that foliar spraying of 37.5–750.0 mg/L Na_2_SeO_3_ could effectively increase the contents of macromolecular bound Se and total Se in pea seedlings, which were 8.75–38.75 times and 6.01–55.65 times higher than those in the control, respectively [38]. In the present study, compared with the control treatment, SeNPs and Na_2_SeO_3_ at the highest concentration (3.0 mmol/L) can increase the Se content in cowpea pods by 538.60% and 800.76%, respectively. In general, the effect of SeNPs on the increase of Se content in cowpea pods is not as good as that of Na_2_SeO_3_, which may be related to their absorption kinetics, different physical and chemical properties, and different interactions with biological molecules [39]. The inward flow mechanism of Na_2_SeO_3_ is different from that of SeNPs: Na_2_SeO_3_ enters plants mainly through the active transport of phosphate transporters, while SeNPs diffuses passively through aquaporins. In wheat and rice, the absorption rate of SeNPs is much lower than that of Na_2_SeO_3_ [39,40]. It is worth noting that the application of 3.0 mmol/L Na_2_SeO_3_ has a toxic effect on cowpea plants, which is manifested by a large number of irregular necrotic spots distributed at the edge of leaves and chlorosis between veins. It is speculated that the leaf spraying treatment with high concentration of Na_2_SeO_3_ may cause the formation of ROS in the mesophyll cells of cowpea, leading to the destruction of cell membrane structure and the decrease of chlorophyll concentration [18]. However, the application of SeNPs with the same concentration did not cause cowpea plants to show obvious symptoms of injury. This phenomenon shows that the plant toxicity of SeNPs is lower than that of selenite, which is consistent with the results observed by Li in the water culture experiment of garlic with different forms of Se [41]. Kovács monitored the Se absorption and accumulation dynamics of *M. sativa* harvested four times in the same growth season. They found that the Se content in the stems and leaves of selenate and selenite treatment groups decreased gradually during the harvest period, and the effect of SeNPs treatment was different from this. In the 10 mg/L SeNPs treatment group, the total Se content of cowpea leaves harvested from the first harvest (14.3 μg/g) to the fourth harvest (37.5 μg/g) gradually increased. In the 50 mg/L SeNPs treatment group, the total Se content decreased in the first three harvests and increased again in the fourth harvest [37].

In this study, cowpea behaves similarly to *M. sativa* under Na_2_SeO_3_ treatment. The total Se content in the pods harvested in the second harvest is lower than that harvested in the first harvest, but the difference is not significant. However, after SeNPs treatment, the Se content in cowpea pods harvested in the second crop was higher than that in the first crop at a high concentration (2.5 mmol/L). It is likely that the difference will be caused by the different oxidation-reduction, assimilation metabolism and transport modes of SeNPs in plant cells.

### 3.3. Effect of Se on Antioxidant Capacity of Cowpea

Higher plants’ antioxidant defense mechanism will be disrupted under adverse environments, resulting in a rise in hydrogen peroxide (H_2_O_2_), superoxide (O_2_^−^) free radicals, MDA, hydroxyl anion (OH^−^), and other reactive oxygen species (ROS) [42,43]. Excessive levels of hydrogen peroxide and superoxide will harm the plasma membrane and chlorophyll. MDA causes cytotoxic cross-linking and polymerization of protein, nucleic acid, and other living components [44].

Appropriate concentration of Se can reduce the active oxygen produced by plant cell membrane, thus promoting the antioxidant metabolism under stress [44]. Se application can also reduce lipid peroxidation by enhancing the activity of antioxidant enzymes against ROS [45,46]. The results showed that the activities of CAT and glutathione reductase (GR) in cowpea root treated with selenite and selenate were significantly increased, and the increase of CAT activity caused by selenite application was more obvious [44]. A total of 5 μmol/L Na_2_SeO_3_ increases the tolerance of cucumber seedlings to oxidative stress caused by water shortage by increasing ascorbic acid peroxidase (APX), POD, SOD and CAT activities, and also inhibits the damage of plasma membrane caused by lipid peroxidation [47]. After *V. radiata* sprouts were sprayed with SeNPs leaves, the activities of SOD and POD showed a trend of rising first and then declining. The maximum activities of two enzymes increased by 108.04% and 59.45%, respectively, compared with the control group [48].

The results of this study show that Se application can prevent the damage caused by peroxidation of cowpea plants to a certain extent. Among them, SeNPs treatment can significantly inhibit or reduce the content of MDA in cowpea pods, and improve the antioxidant capacity of cowpea. Se application also affected the antioxidant enzyme activity of cowpea pods. Two kinds of Se treatments could effectively improve the peroxidase activity in cowpea pods, and higher concentrations of SeNPs could significantly improve the peroxidase activity in cowpea pods. The two Se treatments had little effect on the activity of superoxide dismutase in pods and could effectively maintain the activity of SOD. High concentration of SeNPs can effectively improve the CAT activity of cowpea, and CAT may play an important role in antioxidant defense against high concentration of Se stress.

### 3.4. Effect of Se Treatment on Se Species in Cowpea

The form of Se in plants is the basis for evaluating the effect of Se supplementation. Organic Se forms (Se amino acids) have higher bioavailability than inorganic Se forms (such as selenite and selenate) [29]. The form and content of Se accumulated by plants after applying Se vary with plant species and tissues and organs distributed. For example, SeMet, Se (IV), MeSeCys and γ-Glu-Se-MC can be detected in cowpea grown in natural Se rich soil [49]. Only Se (VI) and SeMet were detected in *M. sativa* after selenate, selenite and SeNPs were applied [37]. After Na_2_SeO_3_ treatment, the main Se compounds in soybean seeds are SeCys2 and SeMet, accounting for about 74%. In addition, there is less MeSeCys (9%). In this study, the main form of Se detected in cowpea pods after treatment with SeNPs and Na_2_SeO_3_ of different concentrations was SeMet, followed by MeSeCys, which accounted for 55–100% and 14.5–32.31%, respectively, in cowpea pods. With the increase of the concentration of exogenous Se, the types of Se compounds detected in the pods gradually increased, but organic Se (especially SeMet) was the main form, and inorganic Se was only detected in the high concentration treatment group. Li et al. also found that when the treatment of SeNPs increased from 10 μmol/L to 30 μmol/L in rice, the form of Se accumulated in the shoot of seedlings increased from one (SeMet) to two species (SeMet and MeSeCys), and SeMet was the main form; In the 10 μmol/L Na_2_SeO_3_ treatment group, the Se forms detected above ground of rice seedlings were SeMet, MeSeCys and Se (VI). It can be seen that the form and content of Se in plant tissues are also closely related to the types of exogenous Se and the amount of Se applied [41].

Se application could maintain or improve the nutritional quality of cowpea, which made it possible to produce Se-rich foods from cowpea. After Se application, the total Se content in cowpea pod, the edible part of cowpea, was significantly increased compared with the control group. Moreover, the Se in the pod was mainly organic Se, which could be better absorbed and utilized by the human body, further indicating the feasibility of Se-rich cowpea production. At present, the research on Se metabolism and transport in leguminous plants is not completely clear. The ABC transporter is a family of transporters with extensive functions. Through the study of the response of ABC transporters to Se in cowpea transcriptome data, we can understand the correlation between ABC transporters and Se transport.

### 3.5. ABC Transporters and Se Absorption

At present, the mechanism of selenite absorption by plants is not clear. Some studies show that phosphate transporters and silicon transporters participate in the absorption of Na_2_SeO_3_ by plants [22,23]. *OsPT2*, a rice phosphorus transporter, can promote the effective absorption of Se in plants. *OsPT2* gene is mainly expressed in roots. After absorbing selenite from the outside and transforming it into organic Se, it is transported to the upper part of roots. Overexpression of *OsPT2* gene can improve the Se tolerance of plants [19]. Analysis of transcriptome data of Se treated *Lolium perenne* showed that 28 ABC transporters in ryegrass might be related to the accumulation of selenides [20]. After sodium selenate hydroponic treatment of tea plants, 16 key genes of ABC transporter were found, 12 of which were up-regulated and 4 were down-regulated. It was speculated that the differential expression of ABC transporter in tea plants might be related to the transport of selenate in roots [50]. After transcriptome sequencing of rice seedling leaves treated with SeNPs and Na_2_SeO_3_, seven differentially expressed ABC transporters were screened, of which five were up-regulated and two were down regulated, indicating that ABC transporters may be involved in Se transport in rice [24].

A total of 131 *ABC* transporters have been identified in the genome of *A. thaliana*, of which 43 are *ABCG* subfamily transporters [51]. A total of 265 *ABC* transporters have been identified in tobacco genome, of which the number of ABCG subfamily transporters is up to 134 [52]. Statistical analysis of cowpea transcriptome showed that there were 107 *ABC* transporter genes, including 27, 14 and 44 ABCB, ABCC and ABCG subfamilies, respectively. It is suggested that ABCB, ABCC and ABCG subfamilies may be the main undertakers of the function of ABC transporter family and play an important role in the transmembrane transport of substances in plants and stress response. The number of ABC subfamily in different plants may be related to the living environment and genetic characteristics of different plants [53]. Compared with animals, the ABC transporters in plants are more diverse in species and function. On the one hand, this may be related to the complex secondary metabolic activities in plants. On the other hand, it plays an important role in plants’ resistance to changes in the external environment [54].

Studies have shown that ABCG transporters play an important role in plant organ development, epidermal cuticle formation, hormone transport, secretion of secondary metabolites, resistance to biotic and abiotic stresses, and plant microbial interaction [55]. The function prediction of *GmABCG40* shows that *GmABCG40* gene may respond to the induction of multiple hormones, especially the transportation of abscisic acid in soybean root, and play an important role in the process of soybean resistance to stress and growth and development [56]. When the expression of *GmABCG1* in roots and pods is high, soybean seeds contain more fat. It can be speculated that the expression of *GmABCG1* may be related to the transportation of lipid substances and seed development [57]. Unlike ABCG, ABCB transporters play an important role in the polar auxin transport, plant resistance to heavy metals and aluminum toxicity, etc. [58]. For example, the TAP type transporter OsABCB27 (OsALS1) located on the vacuole membrane is involved in the response of rice to aluminum stress [59]. Studies have shown that AtABCB14 and AtABCB15 transporters play a role in auxin transport in *A. thaliana*. AtABCB25 is considered to be related to cadmium and lead tolerance. AtABCB27 transporters are located in the vacuole membrane and participate in aluminum metabolism [60,61,62]. It is reported that the transporter OsABCB23 is involved in the transport of iron and sulfur precursors in rice [63].

The number of ABCC subfamilies in plants is only less than that of ABCD and ABCG subfamilies. A total of 15 ABCC subfamilies have been identified in *A. thaliana*, among which AtABCC1 mediates the uptake of radiolabeled folic acid and antifolate [64,65], while AtABCC2 is related to the transport of glucuronic acid conjugates [66]. These two transporters also mediate tolerance to cadmium, mercury and arsenic by transferring phytochelatin into vacuoles [67]. Similarly, in *Sedum alfredii*, the ABCC transporter subfamilies Sa14F190 and Sa18F186 also participate in the tolerance and adaptation of plants to cadmium [68]. In rice, OsABCC1 locates in the vacuole membrane and participates in the accumulation and storage of arsenic in grains [69]. Pan found that plant ABC transporter ABCC8 may act as the plasma membrane transporter of glyphosate, reducing the level of cytoplasmic glyphosate, thus giving plants resistance to glyphosate [70].

In this study, three subfamilies ABCB, ABCC and ABCG obtained from cowpea transcriptome data account for a large proportion in the differential expression of ABC family, indicating that the three subfamilies play a major role in Se stress in leguminous plants. WGCNA analysis showed that the content of organic Se in cowpea materials treated with high concentration of SeNPs was significantly correlated with the expression levels of ABCC11, ABCC13 and ABCC10 transporters. This result suggests that ABCC may be more involved in the transmembrane transport of organic Se in plant cells.

## 4. Materials and Methods

### 4.1. Plant Materials

*Vigna unguiculata* cylindrica was selected as experimental material, and Na_2_SeO_3_ and selenium nanoparticles (SeNPs) were used as exogenous Se to spray cowpea leaves. The experiment site is the nursery garden of Wuhan Light Industry University (the soil type is loam). The sources of applied Se are, respectively, biological SeNPs and Na_2_SeO_3_, SeNPs is provided by Se-rich Microorganism Research Group of Wuhan Polytechnic University, and Na_2_SeO_3_ is purchased from Hubei Cheng Chemical Co., Ltd. (Wuhan, China) [71]. Biosynthesized selenium nanoparticles by *B. subtilis* T5 were characterized systematically using UV-vis spectroscopy, FTIR, Zeta Potential, DLS, and SEM techniques [72].

### 4.2. Treatment of Cowpea with Se

The experiment began sowing on 31 March 2020. The cowpea seeds with full grains were selected and sown on the flat land without weeds, with the plant spacing of 25 cm and the group spacing of 50 cm. On June 3, exogenous Se was sprayed on the leaves at the early flowering stage. Se concentration gradient is set as 0, 0.1, 0.5, 1.0, 2.5, 3.0 mmol/L. Randomized block design was used in the experiment, which was repeated for 3 times (Appendix A). On 11 June 2020, the mature cowpea pods with the same growth status will be picked, washed with ultrapure water, and then subpackaged. They will be placed in the −80 °C refrigerator for the subsequent determination of relevant indicators.

From July to September in 2020, the treatment tests of different harvest time and Se application times were carried out. Sowing on 27 July, applying Se once at the early flowering stage of cowpea (29 August), and sampling twice on 8 September and 25 September, respectively; The second application of Se was carried out in the pod setting period (17 September), and the sampling was carried out on 25 September. The exogenous Se was sprayed on the leaves. The Se source used was SeNPs and Na_2_SeO_3_. The application concentrations were 0 (control), 0.5 mmol/L (low) and 2.5 mmol/L (high), respectively. The experiment was randomized block design and repeated for three times. During the growing period of cowpea, normal water, fertilizer and pest control management shall be carried out. After sampling, dry it, grind it into powder and sieve it for standby.

### 4.3. Chlorophyll, Cellulose, Soluble Sugar and Protein, Free Amino Acid, and Vc Contents

Chlorophyll, cellulose, soluble sugar and protein, free amino acid, and Vc content were determined following the method described by Hou [73]. Briefly, the chlorophyll and carotenoids were extracted with 95% ethanol and measured using an ultraviolet-visible (UV-Vis) spectrophotometer at 665, 649, and 470 nm. Soluble sugar and protein contents were determined by anthrone–sulfuric acid colorimetry at 620 nm and Coomassie bright blue G-250 color rendering at 595 nm, respectively. Total free amino acid content was measured via ninhydrin colorimetry at 570 nm. Vc content was measured using the 2,6-dichloroindophenol titration method.

### 4.4. Determination of Antioxidant Index of Cowpea Pods

The antioxidant index of cowpea pods is determined according to Rao’s method [74]. For the determination of hydrogen peroxide and MDA content, refer to the Kit (A003-1-2, A003-3-5) of Nanjing Jiancheng Biological Engineering Co., Ltd. (Nanjing, China).

The activities of CAT, polyphenol oxidase PPO), peroxidase POD, superoxide dismutase SOD and GR in cowpea pods were measured using commercial enzyme activity assay Kits (Nanjing Jiancheng Technology Co., Ltd., Nanjing, China) in accordance with the manufacturer’ instructions.

### 4.5. Total Se and Se Species Concentrations

The total Se content of cowpea pods was determined using hydride generation atomic fluorescence spectrometry (HG-AFS). Briefly, 0.5 g of dry samples were digested with 10 mL of HNO_3_ and 2 mL of H_2_O_2_ in a microwave digestion system. The digested solutions were added with 5 mL of HCl and then heated until they became clear. The solutions were diluted with water to 10 mL for HG-AFS (AFS8510, Haiguang Instrument, Beijing, China) detection. The working parameters for HG-AFS were as follows: negative high voltage, 340 V; lamp current, 100 mA; atomization temperature, 800 °C; carrier gas flow rate, 500 mL/min; injection volume, 1 mL; KBH4 concentration, 3.5%. Se species analysis was performed via liquid chromatography HG-AFS (LC-AFS8510, Haiguang Instrument, Beijing, China). Five standard substances, namely, SeO3^2−^, SeO4^2−^, SeCys2, SeMet, and MeSeCys, purchased from the National Institute of Metrology (Beijing, China), were employed to plot standard curves. Extraction of Se species was performed by placing 0.2 g of dry samples in centrifuge tubes, immersing the tubes in a water bath with ultrasonication at 70 °C for 1 h, and then centrifuging them at 5000 rpm for 10 min. The supernate was measured via an LC–HG-AFS under the following conditions: mobile phase, 40 mmol/L KH_2_PO_4_ + 20 mmol/L KCl, pH 6.0; flow rate, 1.0 mL/min, chromatographic column, Hamilton PRP-X100 (Hamilton Co., Reno, NV, USA); column temperature, 25 °C; injection volume, 100 μL; cathodic current, 80 mA; carrier gas flow rate, 600 mL/min; and negative high voltage, 350 V. Each treatment included three biological replicates with two technical replicates.

### 4.6. Transcriptome Sequencing and Analysis

Total RNA of four stages were extracted by the Trizol Kit (Promega, Beijing, China) according to the manufacturer’s instructions. The RNA was treated with DNase I (Takara to remove genomic DNA). The RNA integrity and quality were verified by RNase-free agarose gel and NanoDrop 2000 (IMPLEN, Westlake Village, CA, USA).

Total RNA was used as input material for the RNA sample preparations. The clustering of the index-coded samples was performed on a cBot Cluster Generation System using TruSeq PE Cluster Kit v3-cBot-HS (Illumina) according to the manufacturer’s instructions. After cluster generation, the library preparations were sequenced on an Illumina Novaseq platform and 150 bp paired-end reads were generated. Raw data (raw reads) of fastq format were firstly processed through in-house Perl scripts. Q20, Q30 and GC content, the clean data, were calculated. All the downstream analyses were based on the clean data with high quality. Reference genome and gene model annotation files were downloaded from genome website directly. The mapped reads of each sample were assembled by StringTie (v1.3.3b, Johns Hopkins University, Baltimore, MD, USA) in a reference-based approach [75].

Transcriptome assembly was then performed using Trinity software (v2.4, the Broad Institute, Cambridge, MA, USA). Gene function was annotated on the basis of the following databases: the National Center for Biotechnology Information non-redundant protein sequences (Nr), Kyoto Encyclopedia of Genes and Genomes (KEGG), Protein family (Pfam), the evolutionary genealogy of genes: Nonsupervised Orthologous Groups (egg-NOG), Swiss-Prot, and Gene Ontology (GO). Fragments per kilobase of transcript per million fragments (FPKM) was used to represent the expression level of unigenes.

To identify key regulatory genes in the absorption and transport of Se corresponding to SeNPs and Na_2_SeO_3_ treatments, WGCNA was performed by the Novogene online tools (https://magic.novogene.com/customer/ (accessed on 13 September 2022). The module eigengene was defined as the first principal component of a given module, and then used to represent the expression profile of module genes in each sample. The Pearson correlations between the eigengenes of each module and the abundance of flavonoids were performed using R package ggplot2.

### 4.7. Statistical Analysis

Excel 2021 (v2212, Microsoft Corporation, Redmond, WA, USA) and SPSS (v22.0, IBM Corporation, Amonk, NY, USA) were used for experimental data processing. Duncan’s test was used to analyze significant differences (*p* ≤ 0.05). All of the experimental data had three biological replicates. Correlation Network was performed using the OmicStudio tools at https://www.omicstudio.cn/tool (accessed on 21 November 2022). The positive correlation threshold is set to be greater than or equal to 0.5, the negative correlation threshold is set to be less than or equal to −0.5, and the *p*-value threshold is less than 0.5, R version 3.6.1, igraph1.2.6.

## 5. Conclusions

Organic Se is safer than inorganic Se and can be used as a nutritional supplement for people living in Se deficient areas. In the present study, both SeNPs and Na_2_SeO_3_ treatment can increase the soluble sugar content in cowpea pods. SeNPs treatment increased the content of cellulose in cowpea pods. Na_2_SeO_3_ treatment increased the content of Vc in cowpea pods. Se application could effectively alleviate the peroxidation of cowpea plants and enhance the stress resistance of cowpea. Compared with SeNPs, Na_2_SeO_3_ treatment had a better effect on improving the total Se content in cowpea pods. After treatment with SeNPs and Na_2_SeO_3_, the Se form detected in cowpea pods was mainly SeMet, followed by MeSeCys. ABCB, ABCC and ABCG subfamilies play an important role in the absorption and transport of Se in cowpea. The content of organic Se in cowpea treated with high concentration of SeNPs was significantly and positively correlated with the expression level of three transporters ABCC11, ABCC13 and ABCC10, which means that the ABCC subfamily may be more involved in the transmembrane transport of organic Se. In the future, the functions of these three genes need to be verified, such as the transport and absorption of Se in transgenic yeast. Further research will be beneficial to the molecular assisted breeding of Se-rich legume cultivars.

## Figures and Tables

**Figure 1 molecules-28-01398-f001:**
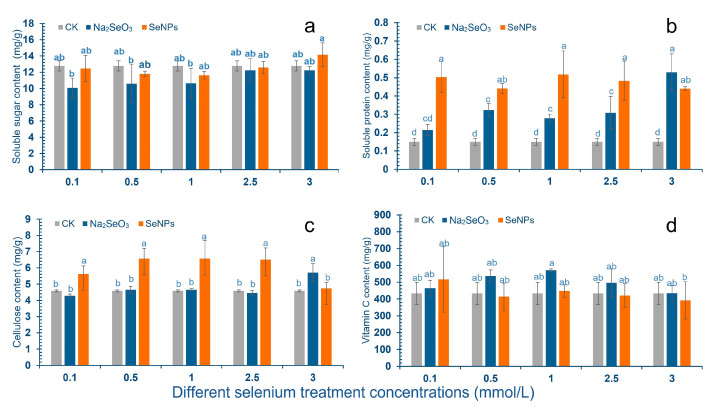
Effects of SeNPs and Na_2_SeO_3_ with different concentrations on the nutrient content of cowpea. The letter a–d represents the different significance under different treatments, *p*-value < 0.05. (**a**) Changes of soluble sugar content in cowpea pods; (**b**) Changes of soluble protein content in cowpea pods; (**c**) Change of cellulose content in pods; (**d**) Changes of Vc content in pods.

**Figure 2 molecules-28-01398-f002:**
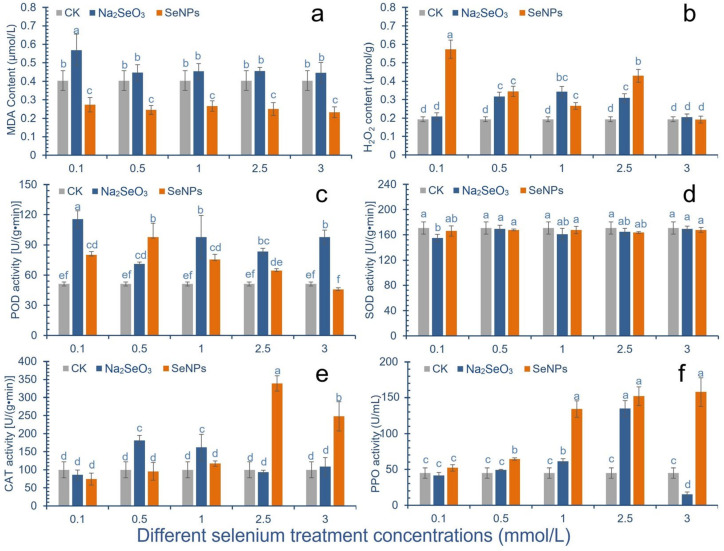
Effect of different Se species treatments on antioxidant activity of cowpea. The letter a–f represents the difference significance under different treatments, *p*-value < 0.05. (**a**) Change of malondialdehyde (MDA) content in pods; (**b**) Change of hydrogen peroxide solution (H_2_O_2_) content in pods; (**c**) Change of peroxidase (POD) activity; (**d**) Changes of superoxide dismutase (SOD) activity; (**e**) Change of catalase (CAT) activity; (**f**) Changes of polyphenol oxidase (PPO) activity.

**Figure 3 molecules-28-01398-f003:**
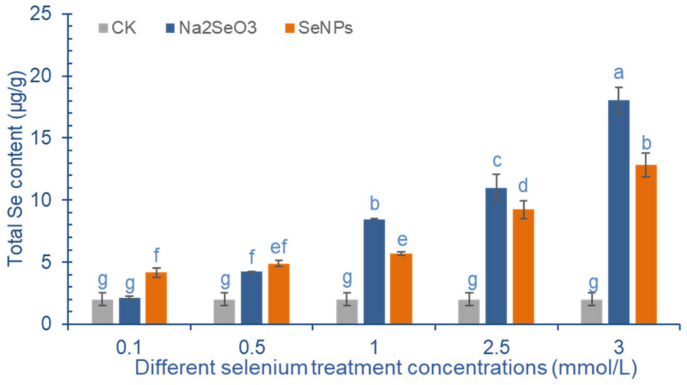
Effect of exogenous Se treatment with different concentrations on total Se content in cowpea. The letter a–g represents the difference significance under different treatments, *p*-value < 0.05.

**Figure 4 molecules-28-01398-f004:**
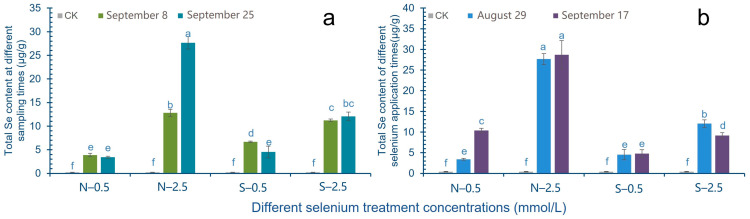
Effects of Se treatment times and picking time on total Se content in cowpea pods. The letter a–f represents the difference significance under different treatments, *p*-value < 0.05. (**a**) Changes of total Se content at different sampling times after Se treatment; (**b**) Change of total Se content in different treatment time.

**Figure 5 molecules-28-01398-f005:**
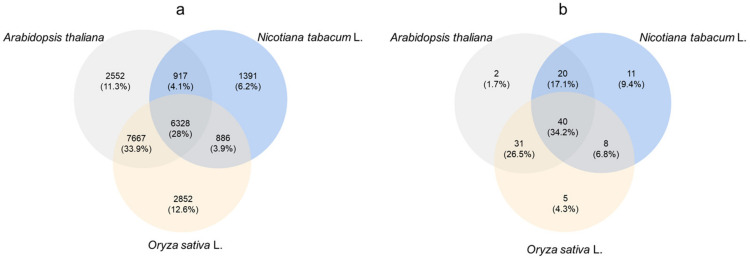
Venn diagram of differentially expressed ABC transporter genes in cowpea transcriptome. (**a**) Annotation of cowpea transcriptome genes with transcriptome data of *A. thaliana*, *Nicotiana tabacum* and *Oryza sativa*; (**b**) The number of ABC transporter genes of cowpea transcriptome annotated to the above three species.

**Figure 6 molecules-28-01398-f006:**
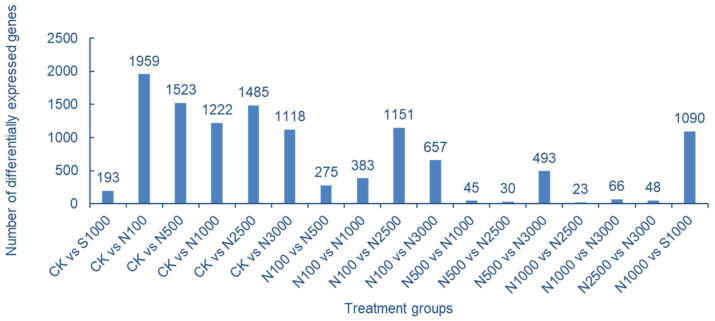
Grouping of DEGs in cowpea transcriptome data.

**Figure 7 molecules-28-01398-f007:**
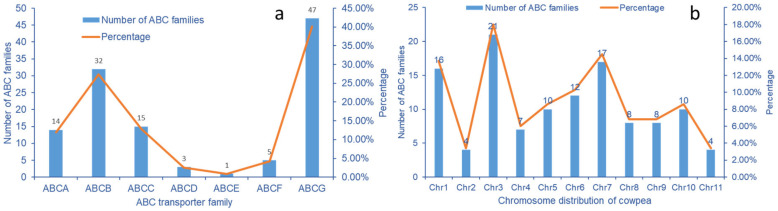
Quantitative distribution of ABC transporter gene subfamilies in *V. unguiculata*. (**a**) Quantitative distribution of ABC transporter family subclasses in cowpea; (**b**) Distribution of ABC transporter subfamily in cowpea genome.

**Figure 8 molecules-28-01398-f008:**
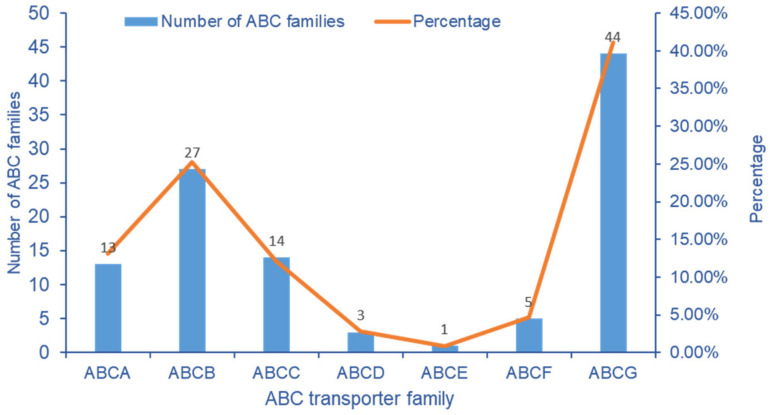
Number distribution of ABC transporter gene family in *V. unguiculata* after Se treatment.

**Figure 9 molecules-28-01398-f009:**
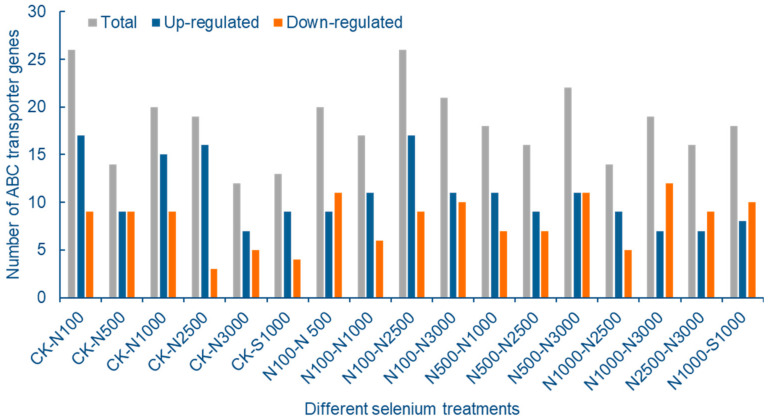
Statistical analysis of differentially expressed cowpea ABC transporter genes under different Se treatments.

**Figure 10 molecules-28-01398-f010:**
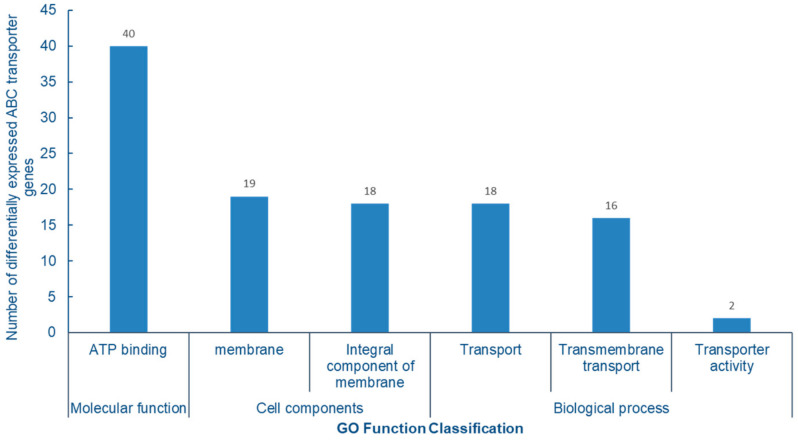
Analysis of GO function enrichment of differentially expressed ABC transporter gene.

**Figure 11 molecules-28-01398-f011:**
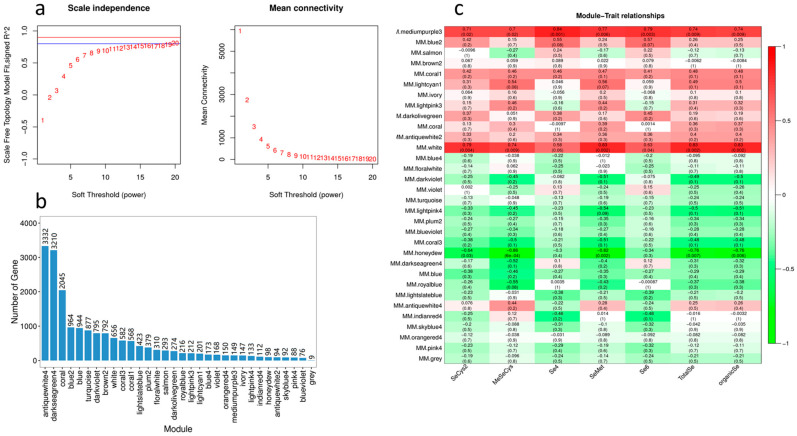
WGCNA analysis of the association between DEGs and different Se forms in cowpea treated with different Se. (**a**) Left: The abscissa represents the power value, the ordinate represents the correlation coefficient, the blue horizontal line represents the correlation coefficient 0.8, and the red horizontal line represents the correlation coefficient 0.9. Right: The abscissa represents the power value, and the ordinate represents the average connectivity of genes; (**b**) Number distribution of genes in different modules, the abscissa represents each module, and the ordinate represents the number of genes; (**c**) Direct correlation between different modules and Se speciation. The abscissa is the character, the ordinate is the module, and Pearson correlation coefficient is used for mapping. Red represents positive correlation and green represents negative correlation. The darker the color is, the stronger the correlation is. The number in the lower brackets represents significant *p* value. The smaller the value is, the stronger the significance is.

**Figure 12 molecules-28-01398-f012:**
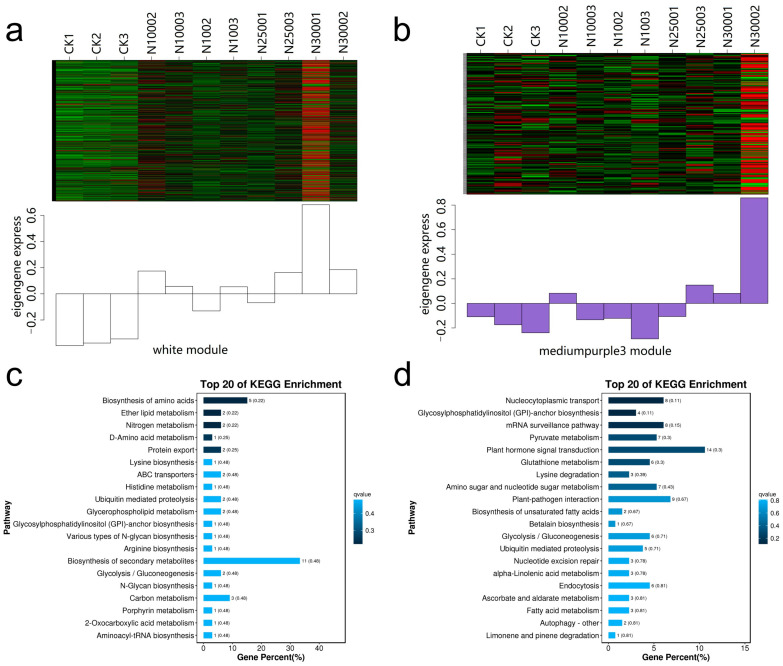
Enrichment analysis of KEGG pathway of white and medium purple3 module genes. (**a**) Heat map of gene expression of white module treated with different Se; (**b**) Heat map of gene expression of medium purple3 module treated with different Se; (**c**) Kegg enrichment bar graph of the first 20 genes with the lowest *p*-value in the white module; (**d**) Kegg enrichment bar graph of the first 20 genes with the lowest *p*-value in the medium purple3 module.

**Table 1 molecules-28-01398-t001:** The content and proportion of Se forms in cowpea pods after SeNPs treatment.

	Content of Different Se Forms				
SeNP Treatment Concentration (mM)	SeCys_2_ (μg/g) DW	MeSeCys (μg/g) DW	SeMet (μg/g) DW	Se^4+^ (μg/g) DW	Se^6+^ (μg/g) DW
CK	-	-	0.1187 ± 0.0059 ^e^ 100.00%	-	-
0.1	-	-	0.2666 ± 0.0028 ^e^ 100.00%	-	-
0.5	-	0.1509 ± 0.0116 ^d^ 28.65%	0.3758 ± 0.0061 ^d^ 71.35%	-	-
1.0	0.0183 ± 0.0057 ^c^ 2.17%	0.2274 ± 0.0195 ^c^ 26.93%	0.5988 ± 0.0084 ^c^ 70.90%	-	-
2.5	0.0587 ± 0.0066 ^b^ 4.45%	0.4261 ± 0.0167 ^b^ 32.31%	0.8338 ± 0.0749 ^b^ 63.24%	-	-
3.0	0.1478 ± 0.0095 ^a^ 6.88%	0.5206 ± 0.0127 ^a^ 24.24%	1.1813 ± 0.0272 ^a^ 55.00%	0.0084 ± 0.0032 ^a^ 0.39%	0.2897 ± 0.0254 ^a^ 13.49%

Note: Small letters a–e indicate the significance analysis of each Se form after treatment with the same Se source and different concentrations, *p*-value < 0.05. The percentage in the table is the proportion of each Se form under the treatment of Se source of this concentration. The short horizontal line represents that the content of the substance is not detected in the sample.

**Table 2 molecules-28-01398-t002:** The content and proportion of Se forms in cowpea pods after Na_2_SeO_3_ treatment.

	Content of Different Se Forms				
Na_2_SeO_3_ Treatment Concentration (mM)	SeCys_2_ (μg/g) DW	MeSeCys (μg/g) DW	SeMet (μg/g) DW	Se^4+^ (μg/g) DW	Se^6+^ (μg/g) DW
CK	-	-	0.1035 ± 0.0059 ^e^ 100.00%	-	-
0.1	-	-	0.1545 ± 0.0119 ^e^ 100.00%	-	-
0.5	0.0136 ± 0.0016 ^c^ 1.21%	0.1672 ± 0.0073 ^c^ 14.82%	0.9473 ± 0.0799 ^c^ 83.97%	-	-
1.0	0.0459 ± 0.0083 ^c^ 3.53%	0.2872 ± 0.0348 ^bc^ 22.09%	0.9671 ± 0.0967 ^c^ 74.38%	-	-
2.5	0.1323 ± 0.0213 ^b^ 4.68%	0.4095 ± 0.0304 ^b^ 14.50%	2.2833 ± 0.1101 ^b^ 80.82%	-	-
3.0	0.2990 ± 0.0121 ^a^ 6.28%	0.8859 ± 0.0517 ^a^ 18.61%	3.4273 ± 0.2747 ^a^ 71.99%	0.1485 ± 0.0412 ^a^ 3.12%	-

Note: Small letters a–e indicate the significance analysis of each Se form after treatment with the same Se source and different concentrations, *p*-value < 0.05. The percentage in the table is the proportion of each Se form under the treatment of Se source of this concentration. The short horizontal line represents that the content of the substance is not detected in the sample.

## Data Availability

Not applicable.

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
