# Peer review of "Effect of Nano-Selenium on Nutritional Quality of Cowpea and Response of ABCC Transporter Family"

_molecules, 2023, doi:10.3390/molecules28031398_

Round 1

Reviewer 1 Report

Title is very long; it can be shortened. Authors used nanopartricle of Se, so it should be mentioned in the title, function of bulk Se and nanoparticles are not same.

In the abstract, introduction of Se is too much, reduce to one line.

POD, PPO and CAT, MDA, SOD. provide full name at the first mentioned place.

Vigna unguiculata scientific names should be italic

Line 43, summarize only important and essential benefits of Se

Line 53 Selenate and Selenite, chemical formula be added here.

Line 93-97, sentence is very long and can be broken into small sentences

Figure 1, soluble sugar, the SD values on the bars are similar, please recheck for confirmation

174, 0.60 μmol/g, 0.35 μmol/g, 1.24 μmol/g. it can be changed to the % increase or decrease

191, SOD in cowpea pods decreased, how much decreased mentioned in %.

In caption of figure 2 (H2O2) , write numerical digits in the subscript or superscript.

347 Fold Changes,….. first letter should be small

423 Nitrogen metabolism, Ether lipid metabolis…. first letter should be small

446 mung bean, provide scientific names

462 drought and 462 high salt environments, why the authors focus on this stresses, please more related references.

471  provide full name of GR 475 APX,

3.3. this heading should be before 2.2, it is important factor.

L. perenne 568, provide full name at the first mentioned place

GmABCG40, OsABCB27 (OsALS1),  gene names should be italic

Material and methods, how Se nanoparticles were prepared, should mention here

Characterization of Nanoparticles should be reported, size, structure, and properties

Conclusions is very long, I suggest to reduce it.

Add the targeted beneficiary audience who will get benefit from this research.

Also, give clear-cut recommendations

Give future perspective regarding this research.

Reviewer 2 Report

  • The manuscript provides enough information and has a certain novelty. The research design is appropriate, the methods are adequately described, and the results are clearly presented. There are several minor problems as follows:

    1. Line from 32 to 35: “Further analysis by WGCNA showed that the content of organic Se in cowpea treated with high concentration of SeNPs was significantly and positively correlated with the expression level of three transporters ABCC11, ABCC13, and ABCC10, which means that ABCC subfamily may be more involved in the transmembrane transport of organic Se in cells”.

    Selenium treatment could enhance the expression of transporter genes, but could not prove that it could transport organic Se, so it needs to be verified by Se uptake in ABCC11-, ABCC13- and ABCC10 -transformed yeast.

    2. The experiments was treated with SeNPs and Na2SeO3, it is necessary to provide the content of Se forms in cowpea pods after Na2SeO3 treatment and the proportion of each Se form.

    3. In Table 1, the content of SeNP form in cowpea pods after SeNPs treatment and the proportion of SeNP form also needed to be provided. SeNP could not be completely converted to other forms.

    4. What are key regulatory genes in the Se-amino acid biosynthesis pathway responding to SeNPs and Na2SeO3 treatments?
